# Impaired Development of Collagen Antibody-Induced Arthritis in Rab44-Deficient Mice

**DOI:** 10.3390/biomedicines12112504

**Published:** 2024-11-01

**Authors:** Yu Yamaguchi, Tomoko Kadowaki, Eiko Sakai, Mayuko Noguromi, Shun Oyakawa, Takayuki Tsukuba

**Affiliations:** 1Department of Dental Pharmacology, Graduate School of Biomedical Sciences, Nagasaki University, Nagasaki 852-8588, Japan; yu-y@nagasaki-u.ac.jp (Y.Y.); eiko-s@nagasaki-u.ac.jp (E.S.); noguromi@nagasaki-u.ac.jp (M.N.); oyakawa@nagasaki-u.ac.jp (S.O.); 2Department of Frontier Oral Science, Graduate School of Biomedical Sciences, Nagasaki University, Nagasaki 852-8588, Japan; tomokok@nagasaki-u.ac.jp; 3Department of Prosthetic Dentistry, Graduate School of Biomedical Sciences, Nagasaki University, Nagasaki 852-8588, Japan

**Keywords:** Rab44, collagen antibody-induced arthritis, immune cells, osteoclasts

## Abstract

Background: Rheumatoid arthritis (RA) is an autoimmune disease characterized by immune cell-mediated joint inflammation and subsequent osteoclast-dependent bone destruction. Collagen antibody-induced arthritis (CAIA) is a useful mouse model for examining the inflammatory mechanisms in human RA. Previously, we identified the novel gene Rab44, which is a member of the large Rab GTPase family and is highly expressed in immune-related cells and osteoclasts. Methods: In this study, we induced CAIA in Rab44-knockout (KO) mice to investigate the effects of Rab44 on inflammation, cell filtration, and bone destruction. Results: Compared with wild-type (WT) mice, Rab44-KO mice showed reduced inflammation in arthritis under CAIA-inducing conditions. Rab44-KO CAIA mice exhibited reduced cell filtration in the radiocarpal joints. Consistent with these findings, Rab44-KO CAIA mice showed decreased mRNA levels of arthritis-related marker genes including genes for inflammation, cartilage turnover, bone formation, and bone absorption markers. Rab44-KO CAIA mice exhibited predominant infiltration of M2-type macrophages at inflammatory sites and reduced bone loss compared to WT CAIA mice. Conclusions: These results indicate that Rab44 deficiency reduces the progression of inflammation in CAIA in mice.

## 1. Introduction

Rheumatoid arthritis (RA) is a chronic autoimmune disease characterized by inflammation of the synovial membrane, leading to joint swelling, pain, and eventual destruction of the cartilage and bone [1]. In the pathogenesis of RA, the misrecognition of self-antigens produces B cell-mediated autoantibodies and induces T cell-mediated stimulation of macrophages and synovial fibroblasts for conversion into tissue-destructive cells [2,3]. Subsequently, the activated cells produce various inflammatory mediators to facilitate osteoclast-dependent bone destruction [4,5]. Because these processes are highly complex, several animal models are available to mimic the pathological conditions of human RA. Collagen antibody-induced arthritis (CAIA) is investigated using a mouse model and is induced by intravenous injection of collagen II antibodies followed by injection with adjuvant booster lipopolysaccharides (LPSs) [6,7]. CAIA is useful not only for examining inflammatory mechanisms in RA but also for screening new therapeutics for regulating inflammatory events in RA [8]. Therefore, performing CAIA experiments in knockout (KO) mice is a powerful tool for analyzing individual gene functions in RA.

Rab GTPases are major regulators of intracellular membrane trafficking [9,10,11]. Rab44 is a member of a group of “large” Rab GTPases and different from conventional “small” Rab GTPases, which include Rab1-43 with low molecular weights of approximately 20–30 kDa [12,13]. Rab44 possesses additional domains: the EF-hand, coiled-coil, and proline-rich domains at the N-terminus and an Rab domain at the C-terminus, with molecular weights of approximately 70–150 kDa [14,15]. Rab44 is upregulated during osteoclast differentiation [14] and is highly expressed in immune-related cells, such as mast cells and granulocyte-lineage cells in the bone marrow [15,16]. Previous studies have reported that Rab44-knockout (Rab44-KO) mice displayed reduced anaphylactic responses and impaired nickel-induced hypersensitivity in vivo [16,17,18]. Moreover, in in vitro experiments, mast cells from Rab44-KO mice exhibited decreased secretion of histamine and lysosomal enzymes compared to cells from wild-type mice [16]. In macrophages, Rab44 knockdown enhances osteoclast differentiation, whereas Rab44 overexpression prevents osteoclast differentiation [14]. Considering that Rab44 has been implicated in the function and differentiation of immune cells and osteoclasts, Rab44 may play an important role in RA development. However, the involvement of Rab44 in the development of RA remains unknown.

In this study, we induced CAIA in Rab44-KO mice to investigate the effects of Rab44 on inflammation, cell filtration, and bone destruction in CAIA.

## 2. Materials and Methods

### 2.1. Antibodies and Reagents

The rabbit polyclonal anti-CD68 antibody (Cat. no. ab125212) was purchased from Abcam (Cambridge, UK). The CD80 polyclonal antibody (Cat. no. bs-2211R) was purchased from Bioss (Woburn, MA, USA). The CD206 rabbit monoclonal antibody (Cat. no. 24595) was obtained from Cell Signaling Technology (Danvers, MA, USA). The ArthritoMAB^TM^ antibody cocktail for C57BL/6 (Cat. no. CIA-MAB-2C) and lipopolysaccharides (LPSs) (Cat. no. MDLPS5-0) were obtained from MDB (Zürich, Switzerland).

### 2.2. Animals

Rab44-KO mice were generated from C57BL/6 background mice using a CRISPR/Cas9-mediated genome editing method described previously [16]. All animal experiments were conducted using age-matched (8–12 weeks old) female wild-type (WT) and Rab44-KO littermate mice. The number of mice used is indicated as “n” in the respective figure legends.

### 2.3. Induction of Collagen Antibody-Induced Arthritis (CAIA)

The age-matched (8–12 weeks old) female mice were intraperitoneally injected with 2 mg of the ArthritoMab arthritis-inducing antibody cocktail on day 0. Three days after the initial administration, the mice were intraperitoneally boosted with 50 μg LPS. The control group was intraperitoneally injected with phosphate-buffered saline (PBS) on days 0 and 3. All mice were monitored for 9 days, and their arthritis development was scored. A maximum of 16 points were assigned, with 4 points per paw (no swelling, 0 points; mild swelling and/or one swollen joint, 1 point; moderate swelling and/or two swollen joints, 2 points; marked swelling and/or all swollen joints, 3 points; and severe swelling with redness and all swollen joints, 4 points). Scoring was conducted according to the method described by Maleitzke et al. [7], with some modifications.

### 2.4. Histopathology

The joint tissues of the forepaws were harvested from the mice. The specimens were fixed in 4% paraformaldehyde and subsequently processed to prepare paraffin-embedded sections. The fixed specimens were decalcified by soaking in formic acid and sodium citrate solution for 7 days. The sections were stained with hematoxylin and eosin (H&E) for inflammation scoring and toluidine blue (TB) for cartilage degradation scoring. TB staining and evaluation methods were based on the method described by Maletzke et al. [19] with some modifications. Briefly, the H&E inflammation scores were defined as follows: 0: normal; 1: mild inflammatory infiltration with no soft tissue edema or synovial lining cell hyperplasia; 2: moderate infiltration with surrounding soft tissue edema and some synovial lining cell hyperplasia; and 3: severe infiltration with marked soft tissue edema and synovial lining cell hyperplasia. The TB staining cartilage scores were defined as follows: 0: normal; 1: mild loss of TB staining; 2: moderate loss of TB staining and cartilage loss; and 3: marked loss of TB staining with marked multifocal cartilage loss.

### 2.5. Immunohistochemistry

The immunohistochemistry analysis was performed according to the method described by Svendsen et al. [20], with some modifications. Briefly, the joint tissues of the forepaws of CAIA mice were fixed with 4% paraformaldehyde and subsequently processed to prepare paraffin-embedded sections. The fixed sections were blocked with 1.0% skim milk in PBS. Samples were incubated with rabbit anti-CD68 (staining for total macrophages), anti-CD80 (staining for M1-type macrophages), and anti-CD206 (staining for M2-type macrophages) primary antibodies, followed by Histofine Simple Stain Mouse MAX-PO (rabbit). The reaction was visualized using 3,3-diaminobenzidine tetrahydrochloride (DAB), and then the sections were counterstained with Harris hematoxylin. Slides were observed under an optical microscope (BZ-X800; KEYENCE, Osaka, Japan). The immunostained samples were analyzed using ImageJ software (ImageJ 1.54d; NIH, Bethesda, MD, USA).

### 2.6. Quantitative Real-Time Polymerase Chain Reaction (qRT-PCR) Analysis

The joints and hind legs were mixed with TRI reagent (Molecular Research Center Inc. Cincinnati, OH, USA). The samples were ultrasonically disrupted for 20 s three times and then centrifuged at 1000 rpm for 10 min. The total RNA was extracted from the supernatants. qRT-PCR analysis was performed as described previously [14]. Reverse transcription was performed using oligo(dT) 15 primer (Promega, Tokyo, Japan) and Revertra Ace (Toyobo, Osaka, Japan). qRT-PCR was conducted using a Quantstudio3 system (Applied Biosystems, Waltham, MA, USA). The cDNA was amplified using Brilliant III Ultra-Fast SYBR Green QPCR Master Mix (Agilent Technologies, Hachioji, Tokyo, Japan). The following primer sets were utilized.
**Gene****Forward Primer****Reverse Primer***Gapdh*AAATGGTGAAGGTCGGTGTGTGAAGGGGTCGTTGATGG*Tnfa*ACGCTGATTTGGTGACCAGG GACCCGTAGGGCGATCAG *Il1b*ACCTAGCTGTCAACGTGTGG TCAAAGCAATGTGCTGGTGC *Cd80*CAAGTTTCCATGTCCAAGGCGGCAAGGCAGCAATACCTTA *Mmp13*GATGGCACTGCTGACATCATTTGGTCCAGGAGGAAAAGC *Il6*CCCCAATTTCCAATGCTCTCCCGCACTAGGTTTGCCGAGTA *Col1a1*TGTTCAGCTTTGTGGACCTCTCAAGCATACCTCGGGTTTC *Col2a1*GGTCCCCCTGGCCTTAGTCCTTGCATGACTCCCATCTG *Sox9*AAGACTCTGGGCAAGCTCTGGGGCTGGTACTTGTAATCGGG *Acan*CAATTACCAGCTGCCCTTCACAGGGAGCTGATCTCGTAGC *Sp7*GCCCCCTGGTGTTCTTCATTCCCATTGGACTTCCCCCTTC *Runx2*GTGGCCACTTACCACAGAGCTGAGGCGATCAGAGAACAAA *Acp5*GGTATGTGCTGGCTGGAAAC ATTTTGAAGCGCAAACGGTA *Ctsk*GTCGTGGAGGCGGCTATATG AGAGTCAATGCCTCCGTTCTG 

The mRNA levels of each transcript were determined relative to the expression levels of the housekeeping gene Gapdh.

### 2.7. Micro-Computed Tomography (μ-CT)

The method used for μ-CT analysis was based on the procedure described by Maleitzke et al. [21], with some modifications. The radiocarpal region with a slice thickness of 2.5 μm was analyzed using a μ-CT SkyScan 1272 scanner (version 1.5, Bruker, Billerica, MA, USA). The analysis region was evaluated within a width of 0.5 mm immediately below the growth plate cartilage. The evaluated bone parameters included the percent bone volume (BV/TV), bone volume fraction (BS/TV), bone surface-to-volume ratio (BS/BV), trabecular number (Tb. N), trabecular thickness (Tb. Th), trabecular separation (Tb. Sp), trabecular bone pattern (Tb. Pf), structural model index (SMI), degree of anisotropy (DA), fractal dimension (FD), connectivity density (Conn. D), and bone mineral density (BMD). NRecon reconstruction software (SkyScan; version 1.7.3.1, Bruker, Billerica, MA, USA) was used for section reconstruction, and 3D image reconstruction was performed using the CTVOX (SkyScan; version 3.3.0 r1403, Bruker) software.

### 2.8. Statistical Analysis

Quantitative data are presented as mean ± standard deviation (SD), except for the arthritis score, which is presented as the standard error of the mean (SEM). The Tukey–Kramer method was used to identify differences between each experimental group when a significant difference (* *p* < 0.05 or ** *p* < 0.01) was determined using one-way analysis of variance (ANOVA).

## 3. Results

### 3.1. Rab44 Deficiency Attenuates Macroscopic Inflammation Induced by Collagen Antibody-Induced Arthritis (CAIA)

To test whether Rab44 is involved in the development of inflammation during arthritis, we induced CAIA in WT and Rab44-KO mice. Figure 1a shows the time course of the CAIA experiments. Upon the CAIA induction, the WT mice displayed severe arthritis in both the forepaws and hindpaws, including the elbow and knee joints (Figure 1b). However, compared to the WT mice, Rab44-KO CAIA mice exhibited impaired arthritis phenotypes, such as edema and redness (Figure 1b). Quantification of arthritis scores revealed that Rab44-KO CAIA mice had an arthritis score approximately half that of WT CAIA mice (Figure 1c). The data indicate that Rab44 deficiency attenuates the macroscopic inflammation induced by CAIA.

### 3.2. Rab44-KO CAIA Mice Exhibit Reduced Cell Filtration in the Radiocarpal Joints

Next, we performed a histopathological analysis of the WT and Rab44-KO CAIA mice. Histopathological analysis of the radiocarpal joints using H&E staining showed significant infiltration with marked soft tissue edema and synovial lining cell hyperplasia in WT CAIA mice (Figure 2a). However, mild inflammatory infiltration with no soft tissue edema or synovial lining cell hyperplasia was observed in Rab44-KO CAIA mice (Figure 2a). Between WT and Rab44-KO control mice, there was no significant morphological difference (Figure 2a). Quantification of the inflammation scores between WT and Rab44-KO CAIA mice revealed significantly reduced scores in Rab44-KO CAIA mice (Figure 2b).

To assess the extent of cartilage degradation, histopathological analysis using toluidine blue staining was performed. WT CAIA mice showed thinning of the cartilage staining layer and decreased staining intensity, particularly in the metacarpophalangeal joint, indicating moderate cartilage degradation (Appendix A). In Rab44-KO CAIA mice, changes in the staining intensity and the size of the stained cartilage layer were less pronounced than those in WT CAIA mice (Appendix A). However, in the quantitative evaluation using the cartilage degradation score, no significant difference was observed between the WT and Rab44-KO CAIA mice (Appendix A).

### 3.3. Rab44-KO CAIA Mice Show Decreased Expression Levels of Arthritis-Related Marker Genes

To investigate the effects of Rab44 on inflammation and the turnover of cartilage and bone, we determined the mRNA levels of marker genes in the ankle joints of WT and Rab44-KO CAIA mice (Figure 3). The mRNA levels of inflammation genes, such as tumor necrosis factor-α (*Tnfa*), interleukin-1β (*Il1b*), interleukin-6 (*Il6*), and CD80 (*Cd80*) in WT and Rab44-KO CAIA joints were all significantly higher than those in the respective WT and Rab44-KO control joints (Figure 3a). Notably, the levels of *Il1b* and *Il6* in KO CAIA mice were significantly lower than those in WT CAIA mice (Figure 3a). However, the mRNA levels of *Tnfa* between the WT and Rab44-KO CAIA mice were comparable (Figure 3a). Interestingly, in Rab44-KO control mice, the mRNA levels of *Il6* were lower but the levels of *Cd80* were higher than those in WT control mice (Figure 3a).

The mRNA levels of the cartilage turnover genes, including collagen type I α 1 chain (*Col1a1*), collagen type II α 1 chain (*Col2a1*), aggrecan (*Acan*) and matrix metalloprotease 13 (*Mmp13*), in WT CAIA mice were significantly higher than those of WT control mice (Figure 3b). However, the levels of *Col1a1* and *Col2a1* in Rab44-KO CAIA mice were similar to those in Rab44-KO control mice, although the levels of Mmp13 in Rab44-KO CAIA mice were significantly higher than in Rab44-KO control mice (Figure 3b). Moreover, the mRNA levels of *Acan* in Rab44-KO CAIA mice were significantly lower than those in Rab44-KO control mice (Figure 3b). Importantly, the mRNA levels of *Col1a1*, *Col2a1*, *Acan* and *Mmp13* in Rab44-KO CAIA mice were significantly lower than those in WT CAIA mice (Figure 3b). However, the levels of SRY-Box transcription factor 9 (*Sox9*) in WT and KO mice were indistinguishable, with or without CAIA induction (Figure 3b).

The mRNA levels of bone formation genes, such as Sp7 transcription factor (Sp7) and RUNX family transcription factor 2 (*Runx2*), in WT CAIA mice were significantly higher than those in WT control mice (Figure 3c). However, both *Sp7* and Runx2 levels were similar between Rab44-KO control and CAIA mice (Figure 3c). Similarly, the levels of bone resorption genes, including tartrate-resistant acid phosphatase type 5 (*Acp5*) and cathepsin K (*Ctsk*) in WT CAIA mice were markedly increased compared to those in the WT control mice (Figure 3d). However, the *Acp5* and *Ctsk* levels in Rab44-KO mice were indistinguishable before and after CAIA induction (Figure 3d).

### 3.4. Rab44-KO CAIA Mice Exhibit Predominant Filtration of M2-Type Macrophages at the Inflammatory Sites

To further investigate the mechanisms of reduced inflammation in Rab44-KO CAIA mice, we examined macrophage subtypes by immunohistochemical analysis (Figure 4). Macrophages are polarized into two subtypes: classically activated macrophages (M1) and alternatively activated macrophages (M2), which have anti-inflammatory activities [22,23]. We selected three different markers: CD68 is a marker for recognizing both M1 and M2 macrophages [24]. CD80 is an M1-type marker [25], and CD206 is an M2-type marker [26]. An immunohistochemical analysis showed that the number of CD68-positive macrophages in Rab44 CAIA mice was apparently lower than that in WT CAIA mice (Figure 4a). The quantitative analysis of CD68-positive macrophages also was consistent with these data (Figure 4b, left panel). Immunohistochemical and quantitative analyses revealed that the number of CD80-positive macrophages (M1) in Rab44-KO CAIA mice was significantly lower than that in WT CAIA mice (Figure 4a,b, middle panel). Conversely, the number of CD206-positive macrophages (M2) in Rab44-KO CAIA mice was significantly higher than that in WT CAIA mice (Figure 4a,b, left panel). These results indicate that Rab44-KO CAIA mice exhibit predominant filtration of M2-type macrophages at the inflammatory sites, although they show reduced filtration of total macrophages.

### 3.5. Rab44-KO CAIA Mice Display Impaired Bone Loss Compared with WT CAIA Mice

Finally, we analyzed the morphological changes in the trabecular bone of the radiocarpal joints of WT and Rab44-KO CAIA mice using micro-computed tomography (μ-CT) (Figure 5a). Upon reconstructing the 3D images and color coding them based on CT values, Rab44-KO CAIA mice showed bone structures with higher CT values than WT CAIA mice (Figure 5b). Radiocarpal trabecular bone volume (BV/TV) fractions were significantly decreased in both types of CAIA mice compared with both groups of control mice (Figure 5c). Moreover, a comparison of various parameters between WT control and WT CAIA mice showed a significant decrease in the bone volume fraction (BS/TV), trabecular number (Tb. N), and fractal dimension (FD), as well as a significant increase in trabecular separation (Tb. Sp), trabecular bone pattern (Tb. Pf), connectivity density (Conn. D), and structure model index (SMI) after CAIA induction (Figure 5c and Appendix A). However, all parameters except for bone surface to volume (BS/BV), Tb. Pf and SMI were indistinguishable between the Rab44-KO CAIA and KO control mice (Figure 5c and Appendix A). In addition, the five indices, including BS/TV, Tb. N, Tb. Sp, FD, and Conn D, indicated that reduced bone loss in Rab44-KO CAIA mice was observed compared with WT CAIA mice (Figure 5c). These results indicate that Rab44-KO CAIA mice display impaired bone loss compared to WT CAIA mice.

## 4. Discussion

In this study, we demonstrated that Rab44 deficiency attenuated CAIA-induced macroscopic inflammation. Histopathological analyses revealed that Rab44-KO CAIA mice exhibited reduced cell filtration in the radiocarpal joints. Real-time PCR analyses indicated that Rab44-KO CAIA mice exhibited decreased expression levels of arthritis-related marker genes. An immunohistochemical analysis showed that Rab44-KO CAIA mice exhibited predominant filtration of M2-type macrophages at inflammatory sites. Micro-CT analyses revealed that Rab44-KO CAIA mice displayed impaired bone loss compared to WT CAIA mice. Thus, Rab44 deficiency likely reduces the progression of inflammation in CAIA in mice.

Consistent with the findings of this study, previous in vivo studies have indicated that Rab44-KO mice display impaired allergic responses, such as anaphylactic responses and nickel-induced hypersensitivity [16,17]. In vitro studies on bone marrow mast cells derived from Rab44-KO mice showed reduced secretion of histamine and β-hexosaminidase compared to those from WT mice. Moreover, Rab44-KO mice exhibited slightly impaired production of TNF-α and interleukin-10 after LPS stimulation [17]. Genetic studies have suggested that human *Rab44* is involved in immune diseases. A study using transcriptome data reported that immune patients with atopy and atopic asthma have differential expression levels of the *Rab44* gene compared to normal individuals [27]. In addition to allergies, Rab44 is likely to be involved in autoimmune diseases. A study using whole-genome sequencing reported that a missense mutation in the Rab44 gene was found in T cells from patients with the autoimmune CD4-lymphoproliferative disease [28]. Considering that Rab44-KO mice exhibit lower levels of CAIA-induced inflammation, it is likely that Rab44 deficiency reduces wide-range immune responses, such as allergies and autoimmune diseases.

The present in vivo study demonstrated that Rab44-KO CAIA mice showed decreased mRNA levels of osteoclasts, such as Acp5 and Ctsk. In contrast, our previous in vitro study indicated that Rab44 knockdown promoted osteoclast differentiation and enhanced bone resorption [14]. The discrepancy in these findings between previous in vitro and present in vivo studies would explain why the upstream effects, including decreased production of cytokines and chemical mediators by Rab44-KO mice, affected the downstream effects, such as decreased levels of the osteoclast markers Acp5 and Ctsk and decreased morphological bone destruction. Consistent with this, previous studies have demonstrated that Rab44-KO mice exhibited reduced inflammatory responses. In fact, Rab44-KO mice exhibited slightly impaired production of TNF-α and interleukin-10 after LPS stimulation [17], and mast cells from Rab44-KO mice showed reduced secretion of histamine and β-hexosaminidase compared to WT mice [16]. Therefore, even if Rab44 deficiency increased osteoclast differentiation and activity in vitro, when the upstream inflammatory response is attenuated in vivo, osteoclast differentiation and bone resorption abilities are also impaired.

Predominant filtration of M2-type macrophages at inflammatory sites was observed in Rab44-KO CAIA mice. Our in vitro studies indicated that Rab44 is upregulated during the early phase of differentiation of M1- and M2-type macrophages [17]. Treatment of human macrophage THP1 cells with LPS and interferon-γ (M1 induction) or IL-4 and IL-13 (M2 induction) increases the expression levels of Rab44 in the early phase [17]. However, the detailed mechanisms underlying the involvement of Rab44 in the polarization of M1 and M2 macrophages remain unclear. Therefore, Rab44 may be involved in immune cell differentiation. This hypothesis is based on several findings. First, Rab44 expression levels are decreased during the differentiation of mature immune cells such as macrophages, neutrophils, and dendritic cells, whereas Rab44 is highly expressed in undifferentiated hematopoietic cells of the bone marrow [15]. Alternatively, after treatment with LPS, Rab44 is partially translocated into early endosomes and the plasma membrane, whereas Rab44 is mainly localized in lysosomes of macrophages [15]. Thus, it is important to determine the mechanisms underlying Rab44 involvement in the polarization of M1 and M2 macrophages in future studies.

Our results provide new insights into the understanding of inflammation, cell filtration, and bone destruction in Rab44-KO mice. However, the lack of osteoclast examination and circulating cytokine examination is a limitation. Moreover, the involvement of Rab44 in the pathogenesis of RA in humans remains to be determined. Given that the inhibition of Rab44 attenuates RA pathogenesis in humans, the development of an Rab44 inhibitor may represent a potential new therapeutic agent.

## 5. Conclusions

In conclusion, Rab44-KO CAIA mice exhibited reduced inflammation and total macrophage filtration but increased M2-type macrophage filtration. Consistent with these findings, Rab44-KO CAIA mice showed decreased expression levels of arthritis-related marker genes, and impaired bone loss compared to WT CAIA mice. Thus, it is likely that Rab44 deficiency reduces the progression of inflammation in CAIA in mice.

## Figures and Tables

**Figure 1 biomedicines-12-02504-f001:**
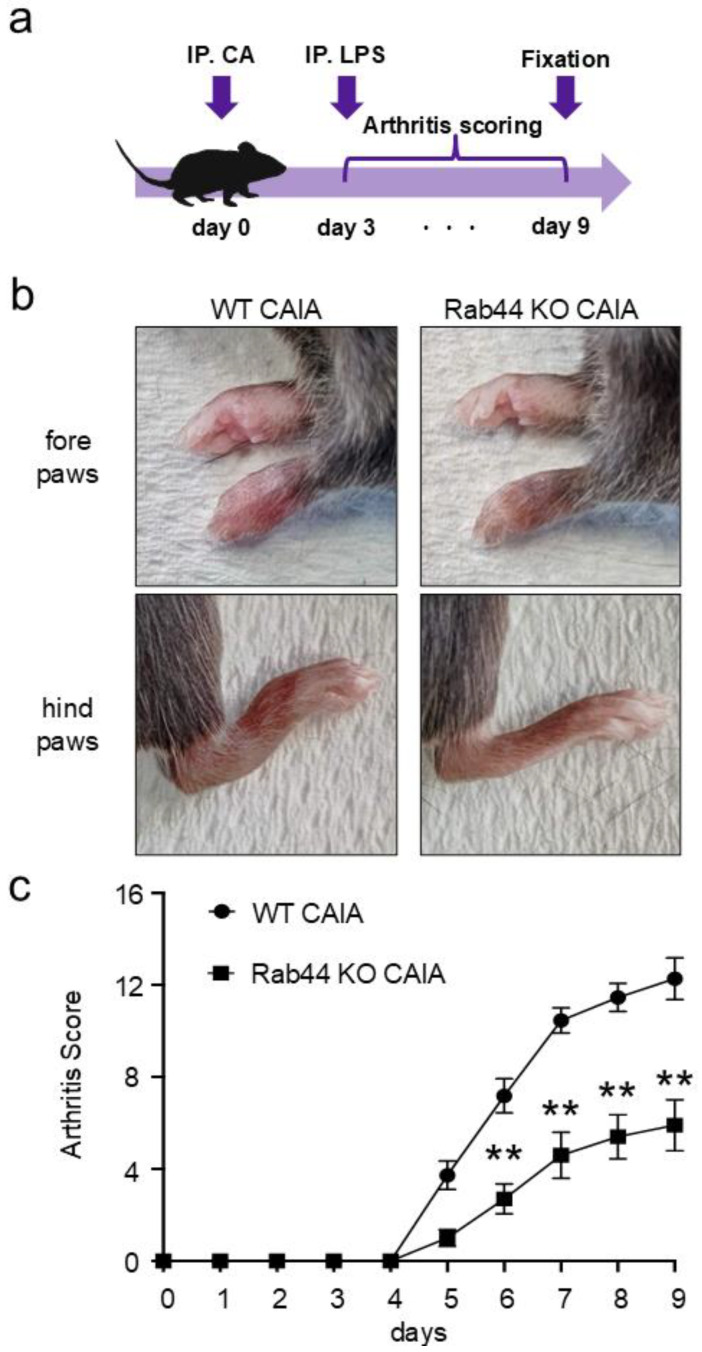
Development of collagen antibody-induced arthritis (CAIA). Wild-type (WT) and Rab44-knockout (KO) mice were intraperitoneally injected with collagen II antibody cocktail (2 mg) on day 0; subsequently, they were intraperitoneally injected with lipopolysaccharide (LPS) (50 μg) on day 3. The mice were monitored for arthritis development over 0–9 days by daily scoring. (**a**) Schematic diagram showing the time course for creating CAIA mice. (**b**) Photographs of the forepaws and hindpaws of WT and Rab44-KO CAIA mouse on day 9. (**c**) Arthritis scores of the WT and Rab44-KO CAIA mice. Scoring was calculated with a maximum of 16 points, with 4 points assigned per paw (no swelling, 0 points; mild swelling and/or one swollen joint, 1 point; moderate swelling and/or two swollen joints, 2 points; marked swelling and/or redness in all swollen joints, 3 points; severe swelling with redness in all swollen joints, 4 points). WT: *n* = 11, Rab44-KO: *n* = 10, ** *p* < 0.01.

**Figure 2 biomedicines-12-02504-f002:**
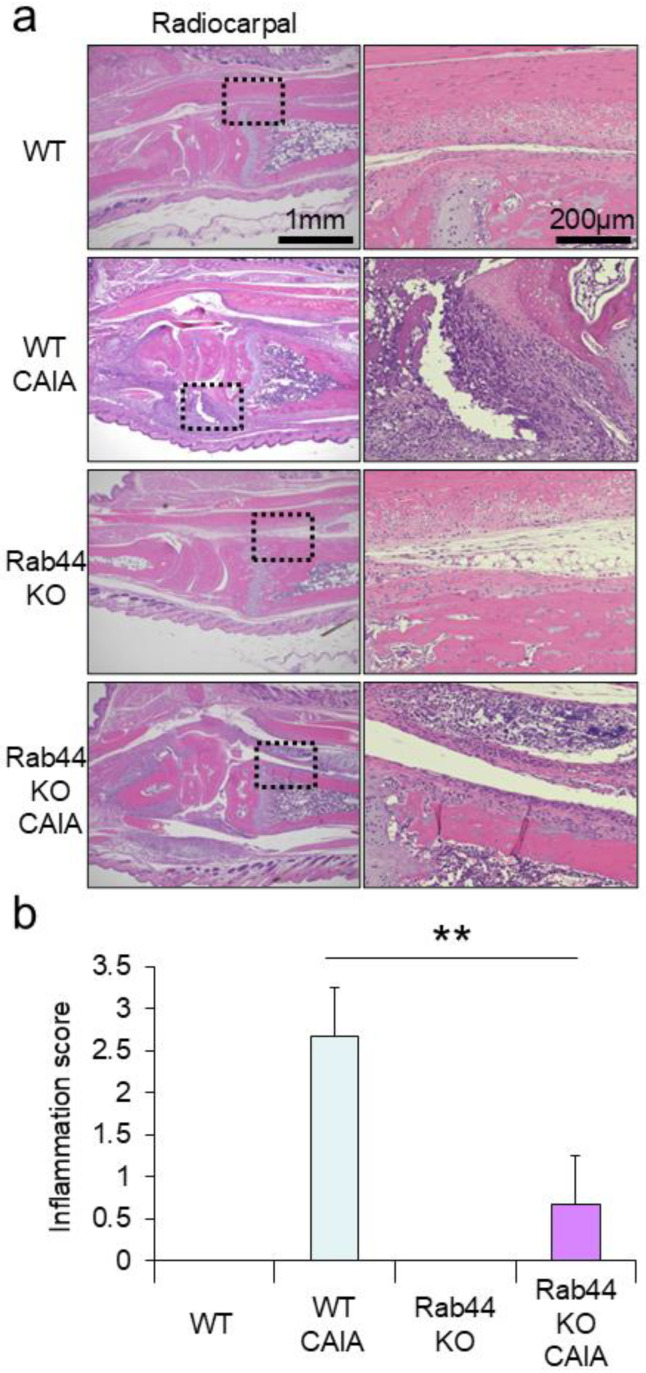
Histological analysis of radiocarpal joints of mice. (**a**) Hematoxylin and eosin (H&E) staining of radiocarpal joints of wild-type (WT) control, WT collagen antibody-induced arthritis (CAIA), Rab44-knockout (KO) control, and Rab44-KO CAIA mice. The dotted lines in the left panel are shown enlarged in the right panel. (**b**) Inflammation scores were counted with H&E staining histological data. Score 0, normal; Score1, mild cell infiltration with no soft tissue edema or synovial lining cell hyperplasia; Score 2, moderate cell infiltration with surrounding soft tissue edema and some synovial lining cell hyperplasia; Score 3, severe cell infiltration with marked soft tissue edema and synovial lining cell hyperplasia (*n* = 3, ** *p* < 0.01).

**Figure 3 biomedicines-12-02504-f003:**
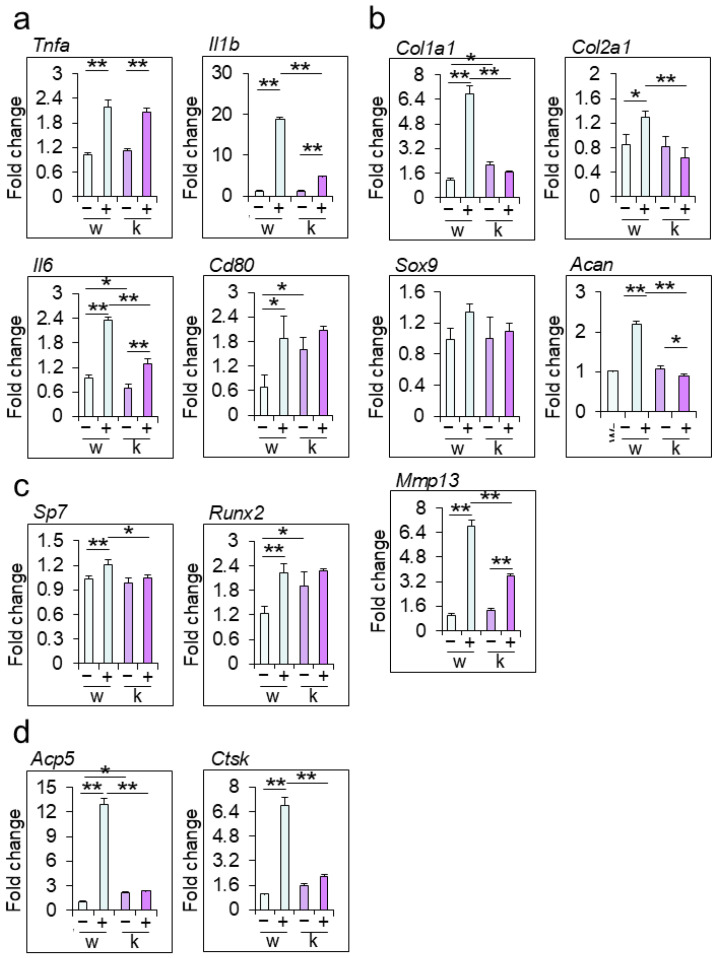
mRNA levels of arthritis marker genes in the ankle joints of mice. mRNA was extracted from the ankle joints of wild-type (WT) control (w−), WT collagen antibody-induced arthritis (CAIA) (w+), Rab44-knockout (KO) control (k−), and Rab44-KO CAIA mice (k+). Subsequently, qRT-PCR was performed. (**a**) Inflammatory markers, tumor necrosis factor-α (*Tnfa*), interleukin-1β (*Il1b*), interleukin-6 (Il6), and CD80 (*Cd80*). (**b**) Cartilage turnover markers collagen type Iα 1 chain (*Col1a1*), collagen type IIα 1 chain (*Col2a1*), SRY-Box transcription factor 9 (*Sox9*), aggrecan (*Acan*) and matrix metalloprotease 13 (*Mmp13*). (**c**) Bone formation markers, Sp7 transcription factor (*Sp7*) and RUNX family transcription factor 2 (*Runx2*). (**d**) Bone resorption markers, tartrate-resistant acid phosphatase type 5 (Acp5) and cathepsin K (*Ctsk*). (* *p* < 0.05, ** *p* < 0.01).

**Figure 4 biomedicines-12-02504-f004:**
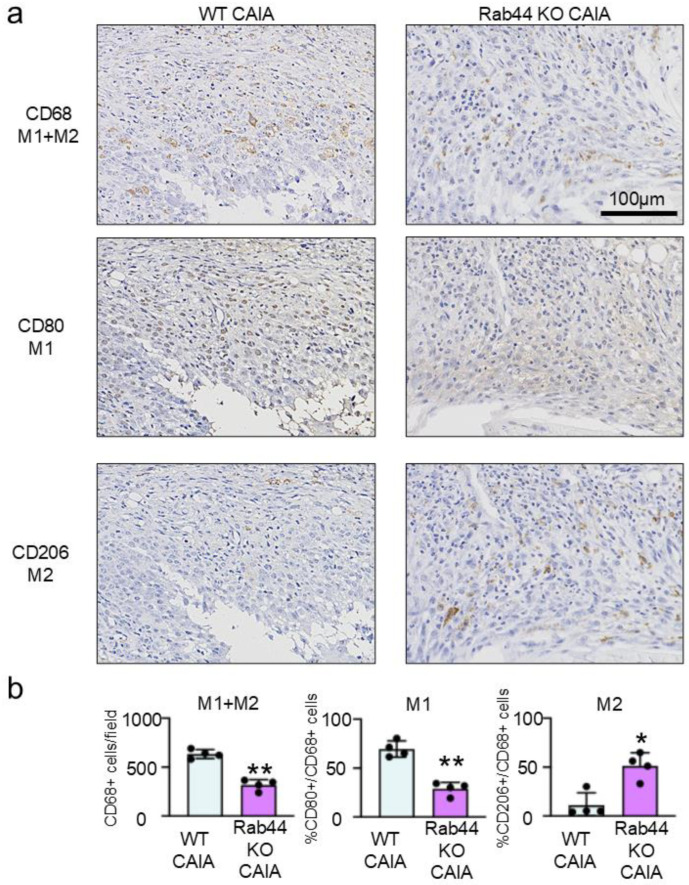
Immunohistochemical analysis of M1- and/or M2-type macrophages in the joint of collagen antibody-induced arthritis (CAIA) mice. (**a**) The fixed sections of wild-type (WT) and Rab44-knockout (KO) CAIA mice were blocked with 1.0% skim milk in PBS. The samples were incubated with rabbit anti-CD68 IgG (total macrophage marker), CD80 IgG (M1-type macrophage marker) and CD206 IgG (M2-type macrophage marker) as the primary antibodies, followed by an HRP/DAB detection method. Samples were observed under an optical microscope. Bars: 100 μm. (**b**) Quantitative analysis of the serial sections with the number of CD68-positive cells visualized in a certain field (left panel), the percentage of the number of CD80-positive cells in the number of CD68-positive cells (middle panel), and the percentage of the number of CD206-positive cells in the number of CD68-positive cells (right panel). Data are represented as the mean ± S.E. of results from four independent experiments. Asterisks indicate statistical significance compared to the control; (*n* = 4, * *p* < 0.05, ** *p* < 0.01).

**Figure 5 biomedicines-12-02504-f005:**
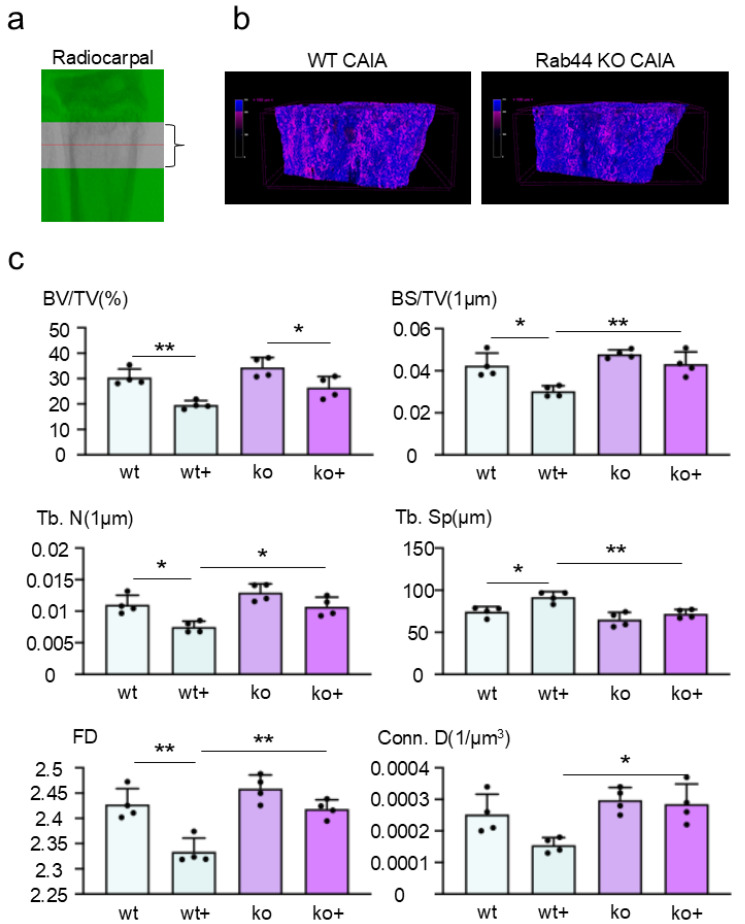
Micro-computed tomography (μ-CT) analysis of the trabecular bone of the radiocarpal joints of mice. The trabecular bones of wild-type (WT) control (w−), WT collagen antibody-induced arthritis (CAIA) (w+), Rab44-knockout (KO) control (k−), and Rab44-KO CAIA mice (k+) were analyzed using μ-CT. (**a**) The analyzed area of the trabecular bone by μ-CT. Distal radius 0.5 mm length. (**b**) Representative 3D μ-CT images of trabecular bone at the distal radius, color coded according to CT value. Areas where the CT values were lower than others are shown in reddish-purple. (**c**) Results showing percent bone volume (BV/TV), bone volume fraction (BS/TV), trabecular number (Tb. N), trabecular separation (Tb. Sp) fractal dimension (FD), and connectivity density (Conn. D) of the trabecular bones at the distal end of the radius in wild-type (WT) control (w−), WT collagen antibody-induced arthritis (CAIA) (w+), Rab44-knockout (KO) control (k−), and Rab44-KO CAIA mice (k+), (*n* = 4, * *p* < 0.05, ** *p* < 0.01).

## Data Availability

The data used to support the findings of this study are available from the corresponding author upon request.

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
