# Peer review of "Impaired Development of Collagen Antibody-Induced Arthritis in Rab44-Deficient Mice"

_biomedicines, 2024, doi:10.3390/biomedicines12112504_

Round 1
Reviewer 1 Report
Comments and Suggestions for Authors
I thank the editors for choosing me to review this manuscript. The article is well written and of considerable interest. Some small corrections are needed which I will list below:
Methods:
- specify the type of study and add the name and number of the ethics committee
Discussion:
- add the limitations of your study and provide suggestions for future research
Author Response
Reviewer#1
Comments and Suggestions for Authors
I thank the editors for choosing me to review this manuscript. The article is well written and of considerable interest. Some small corrections are needed which I will list below:
Methods:
- specify the type of study and add the name and number of the ethics committee
Answer: First of all, we appreciate positive responses for our paper. We added description of specify the type of study and add the name and number of the ethics committee. Line 78-80. The experiments in this study induced an animal biosafety level designation and physical containment for genetically modified animals. (Title: Analysis of lysosomal functions in animal cells; permit number: 2107211733).
Discussion:
- add the limitations of your study and provide suggestions for future research
Answer: Thank you for your important comments. According to the reviewer’s suggestion, we added a description as follows. Line 367-372: Our results provide new insights into the understanding of inflammation, cell filtration, and bone destruction in Rab44-KO mice. However, the lack of osteoclast examination and circulating cytokine examinations is a limitation. Moreover, the involvement of Rab44 in the pathogenesis of RA in humans remains to be determined. Given that the inhibition of Rab44 attenuates RA pathogenesis in humans, the development of a Rab44 inhibitor may represent a potential new therapeutic agent.
Reviewer 2 Report
Comments and Suggestions for Authors
Review of biomedicines-3239362: Impaired development of collagen antibody-induced arthritis in Rab44-deficient mice
Rheumatoid arthritis (RA) is a common autoimmune disease, but there are currently no effective preventive or curative treatments. The authors investigated the effect of Rab44 on the collagen antibody-induced arthritis (CAIA) model, using Rab44 knockout (KO) mice to assess its impact on inflammation, cell infiltration, and bone destruction. Their findings revealed that Rab44-KO CAIA mice exhibited predominant infiltration of M2-type macrophages at inflammatory sites and reduced bone loss compared to wild-type (WT) CAIA mice. These results suggest that Rab44 deficiency mitigates the progression of inflammation in CAIA in mice. Therefore, this manuscript offers valuable insights and is recommended for publication in Biomedicines, pending revisions to address the following edits/comments:
· In Figure 4A, appropriate controls for the immunohistochemical stainings are missing. Please provide control IgG staining for each antibody and section.
A few suggestions the authors might want to consider:
· In the results section, especially in Figures 3, 4, 5, and 7, including individual data points (data dots) within the bars would enhance the visualization of data distribution and variability, providing a more comprehensive understanding of the results.
· In the figure legends, the authors used bold font in the title of the Figure 3 legend, but this is inconsistent with the formatting of other figure legends.
· In the legend for Figure 5, line 294, there is an extra space after the word "mice."
Author Response
Reviewer 2
Comments and Suggestions for Authors
Review of biomedicines-3239362: Impaired development of collagen antibody-induced arthritis in Rab44-deficient mice
Rheumatoid arthritis (RA) is a common autoimmune disease, but there are currently no effective preventive or curative treatments. The authors investigated the effect of Rab44 on the collagen antibody-induced arthritis (CAIA) model, using Rab44 knockout (KO) mice to assess its impact on inflammation, cell infiltration, and bone destruction. Their findings revealed that Rab44-KO CAIA mice exhibited predominant infiltration of M2-type macrophages at inflammatory sites and reduced bone loss compared to wild-type (WT) CAIA mice. These results suggest that Rab44 deficiency mitigates the progression of inflammation in CAIA in mice. Therefore, this manuscript offers valuable insights and is recommended for publication in Biomedicines, pending revisions to address the following edits/comments:
In Figure 4A, appropriate controls for the immunohistochemical stainings are missing. Please provide control IgG staining for each antibody and section.
Answer: Thank you for your important comments. Previously, we provided the data of control IgG staining in several sections. The data are published in the paper (Tokuhisa et al. Sci Rep. 2020 Jul 1;10(1):10728). For your information, Fig 2 shows the staining patterns of control IgG in the bone marrow, spleen, thymus, lung and skin. We used the same IgG for the experiments in Figure 4A.
A few suggestions the authors might want to consider:
In the results section, especially in Figures 3, 4, 5, and 7, including individual data points (data dots) within the bars would enhance the visualization of data distribution and variability, providing a more comprehensive understanding of the results.
Answer: According to the reviewer’s suggestion, we show the individual data points (data dots) within the bars in Figure 4 and Figure 5. However, Figure 3 is the data of RT-PCR, relative quantitation analysis. Due to the characteristics of the analysis, we think it's not appropriate to display the data in dots. Please understand the conditions.
In the figure legends, the authors used bold font in the title of the Figure 3 legend, but this is inconsistent with the formatting of other figure legends.
Answer: This comment is also quite right. We are very sorry, but we showed plain font in several figure legends by mistake. We changed bold font in all figure legends.
In the legend for Figure 5, line 294, there is an extra space after the word "mice."
Answer: We are sorry for our careless miss. We deleted an extra space after the word "mice." in Figure 5, line 294 (shifted to 302).
Reviewer 3 Report
Comments and Suggestions for Authors
The animal study aimed to determine the role of Rab44 in an inflammation model due to rheumatoid arthritis. The results showed that Rab44 knockout significantly reduced cartilage deterioration, bone loss and inflammation in collagen-induced osteoarthritis model. The manuscript has merits to be published and some of the gaps yet to bridged can be mentioned in the limitation paragraph.
Section 2.3: provide a citation for the arthritis score.
Section 2.4: describe briefly the inflammation and cartilage degradation scoring for the benefit of the readers, although the authors have provided the citations for both methods. Who performed the scoring? Was the person blinded to the grouping?
Section 2.3 and 2.4: Do you decalcify the bone before histological processing? If yes, what reagent has been used and for how long.
Section 2.6: How do you process the joint and rear paws for mRNA extraction?
Section 2.6: Suggest to put the nucleotide sequence using a table.
Section 2.8: Please indicate the normality test used and results. Are you using factorial ANOVA or one-way ANOVA?
Discussion: The lack of osteoclast examination and circulating cytokine examination can be mentioned as the limitation.
Comments on the Quality of English LanguageMinor edits only
Author Response
Reviewer 3
Comments and Suggestions for Authors
The animal study aimed to determine the role of Rab44 in an inflammation model due to rheumatoid arthritis. The results showed that Rab44 knockout significantly reduced cartilage deterioration, bone loss and inflammation in collagen-induced osteoarthritis model. The manuscript has merits to be published and some of the gaps yet to bridged can be mentioned in the limitation paragraph.
Section 2.3: provide a citation for the arthritis score.
Answer: Thank you for your important comments. We added the reference based on the arthritis score as follows. Line 90-91: “Scoring was conducted according to the method described by Maleitzke et al. [7] with some modifications”.
Section 2.4: describe briefly the inflammation and cartilage degradation scoring for the benefit of the readers, although the authors have provided the citations for both methods. Who performed the scoring? Was the person blinded to the grouping?
Answer: Thank you for your good suggestion. We described the H&E for inflammation score and TB staining score as follows. Lines 99-105 “Briefly, the H&E for inflammation scores were defined as follows: 0: normal; 1: mild in-flammatory infiltration with no soft tissue edema or synovial lining cell hyperplasia; 2: moderate infiltration with surrounding soft tissue edema and some synovial lining cell hyperplasia; and 3: severe infiltration with marked soft tissue edema and synovial lining cell hyperplasia. The TB staining cartilage scores were defined as follows: 0: normal; 1: mild loss of TB-staining; 2: moderate loss of TB-staining and cartilage loss; and 3: marked loss of TB-staining with marked multifocal cartilage loss.” Moreover, the scoring was mainly conducted by the first author Dr. Yu Yamaguchi. The data was checked as a blind test by the second author Tomoko Kadowaki.
Section 2.3 and 2.4: Do you decalcify the bone before histological processing? If yes, what reagent has been used and for how long.
Answer: We appreciate the important comment. The fixed specimen was decalcified by soaking in formic acid and sodium citrate solution for 7 days.
Section 2.6: How do you process the joint and rear paws for mRNA extraction?
Answer: Thank you for your good suggestion. The joints and hind legs were mixed with TRI reagent (Molecular Research Center Inc. Cincinnati, OH, USA). The samples were ultrasonically disrupted for 20s three times and then centrifuged at 1,000 rpm for 10 min. The total RNA was extracted from the supernatants.
Section 2.6: Suggest to put the nucleotide sequence using a table.
Answer: According to reviewer’s suggestion, we show the table containing the nucleotide sequence, lines 128-141.
Section 2.8: Please indicate the normality test used and results. Are you using factorial ANOVA or one-way ANOVA?
Answer: Thank you for your comment. We used one-way ANOVA. Therefore, we added the description, such as “The Tukey-Kramer method was used to identify differences between each experimental group when a significant difference (*P < 0.05 or **P < 0.01) was determined using one way- analysis of variance (ANOVA).”
Discussion: The lack of osteoclast examination and circulating cytokine examination can be mentioned as the limitation.
Answer: Thank you for your important comments. We added the description “The lack of osteoclast examination and circulating cytokine examination can be mentioned as the limitation.”